# Physical Activity and Quality of Life among Patients with Cystic Fibrosis

**DOI:** 10.3390/children9111665

**Published:** 2022-10-31

**Authors:** Stavros Giannakoulakos, Maria Gioulvanidou, Evangelia Kouidi, Pauline Peftoulidou, Syrmo Styliani Kyrvasili, Parthena Savvidou, Asterios Deligiannis, John Tsanakas, Elpis Hatziagorou

**Affiliations:** 1Sports Medicine Laboratory, Aristotle University of Thessaloniki, 57001 Thermi, Greece; 2Pediatric Respiratory Unit, 3rd Paediatric Clinic, Aristotle University of Thessaloniki, Hippokration General Hospital of Thessaloniki, 54642 Thessaloniki, Greece

**Keywords:** physical activity, cystic fibrosis, quality of Life, DISABKIDS questionnaire

## Abstract

Background: Physical activity (PA) improves exercise capacity, slows the decline in lung function, and enhances Quality of Life (QoL) in patients with cystic fibrosis (pwCF). Objectives: The study aimed to evaluate PA and QoL among children with CF compared to healthy controls; the secondary aim was to assess the correlation between PA, QoL, and lung function (FEV1). Methods: Forty-five children and adolescents with CF and 45 age-matched controls completed two self-administered validated questionnaires: The Godin Leisure-Time Exercise Questionnaire (GLTEQ) and the DISABKIDS for QoL. Moreover, pwCF performed spirometry and multiple breath washout tests (MBW). In addition, weight, height, and BMI were recorded. The Godin Leisure-Time Exercise Questionnaire was used to evaluate physical activity; QOL was assessed using the DISABKIDS Questionnaire. The correlation of PA with QOL was assessed as well. Results: Mean age of the CF population was 13.22 (±4.6) years, mean BMI 19.58 (±4.1) kg/m^2^, mean FEV1% 91.15 ± 20.46%, and mean LCI 10.68 ± 4.08. 68% of the CF group were active, 27% were medium active, 5% were sedentary, while 83% of the control group were active and 17% were medium active. PwCF with higher PA scores showed significantly higher emotional health (r^2^: 0.414, *p*: 0.006) and total QOL score (r^2^: 0.372; *p*: 0.014). The PA score showed no significant correlation with FEV1% or LCI. Conclusions: The children with CF showed satisfactory PA levels, which positively correlated to their QoL. More research is needed on the effect of increased levels of habitual physical activity to establish the decline in pulmonary function among pwCF.

## 1. Introduction

Cystic fibrosis (CF) is the most common autosomal inherited disease in the Caucasian population, affecting 70,000 people worldwide [1]. Due to the complexity of the disease, people with CF (pwCF) may experience symptoms in various systems, such as the respiratory, gastrointestinal, and musculoskeletal [2]. Respiratory dysfunction and recurrent infections lead to lung disease progression, which is the most crucial feature of this systematic disease [3]. Over the last three decades, a continuous increase in life expectancy has been attributed to the early diagnosis of the disease, the timely implementation of appropriate treatment, and novel available treatment options [4]. Besides inhaled medications and the performance of airway clearance techniques (ACTs), regular exercise is recently recommended as an essential component of standard CF care [5].

Physical activity (PA) is associated with various benefits among pwCF, including improvements in respiratory disease [6], mucociliary clearance [7], bone mineral density [8], and fewer hospitalizations [9]. Physical activity is one of the cornerstones of children’s development and socialization, yet children with CF often show less strenuous PA than age-matched controls [10]. Higher levels of PA are associated with improved exercise capacity, prognosis, and quality of life, while a sedentary lifestyle and lower exercise capacity have been associated with higher mortality rates and deteriorated pulmonary function [11]. In addition, it has been suggested that aerobic and anaerobic capacities in most pwCF are reduced relative to their healthy counterparts [12,13]. The reduction in levels of PA is attributed to various potential barriers, including impaired lung function, physical symptoms (breathlessness, cough, fatigue), and a high treatment burden [14]. It has been established that children and adolescents with cystic fibrosis exhibit exercise intolerance [15]. This significantly affects their welfare and quality of life and reduces their opportunities to participate in sports. Except for the lung function decline in pwCF, a poor nutritional status due to digestive impairments and muscle weakness are considered additional contributing factors to their exercise intolerance [16]. An improved compulsory exercise program for pwCF may improve the quality of patients’ clinical care [17].

Increasing physical activity and participation in exercise programs have significant benefits for CF patients. It has been observed that with regular exercise programs 3–5 times a week, with proper warm-up and recovery, CF patients exhibit significant clinical improvement [18]. If adapted to each CF patient’s needs, participation in exercise programs can be safe and therapeutic. However, there may be some complications during exercise, especially in the case of patients with critical severity. The most common complications are violent coughing attacks, exercise-induced asthma, and weight loss. The latter is considered a rather critical condition in young CF patients. For this reason, it is necessary to adopt the patients’ dietary habits, depending on their personal needs, habits, and intensity of disease symptoms [19,20].

In recent years, the interest in CF care has shifted towards health-related Quality of Life measurements, and the results have been proven increasingly valuable [21]. CF adolescents and adults experience a gradual reduction in QoL with increasing disease severity [22]. Furthermore, HRQoL has been used as a predictive survival factor in pwCF [21]. These findings further support the conviction that interventions to improve and maintain related CF adults’ quality of life are significant and should receive greater emphasis since the subject’s condition deteriorates [23,24]. FEV1%, age, sex, body mass index, and pulmonary exacerbations contribute to health-related QoL in adolescents and adults with CF.

The study aimed to evaluate PA and QoL among children with cystic fibrosis compared to healthy controls; the secondary aim was to evaluate the correlation between PA, QoL, and lung function (FEV1).

## 2. Methods

### 2.1. Study Design

The study population included 45 children and young adults aged 5–26 years with a confirmed diagnosis of CF (study group) monitored in a referral CF Center in Northern Greece. Moreover, 45 healthy individuals matched for sex and age served as a control group. The Outpatient CF Clinic collected demographics and patient characteristics over 12 months. All children completed two questionnaires. Moreover, pwCF underwent spirometry and Multiple-Breath Washout (MBW) testing. Informed consent was obtained from all subjects involved in the study. The Institutional Review Board of Aristotle University of Thessaloniki Medical School approved the study. The time needed to complete the questionnaires was approximately 15 to 20 min.

### 2.2. Anthropometric Measurements

Height and weight were measured in light clothing, and Body Mass Index (BMI) was calculated.

### 2.3. Questionnaires

The Godin Leisure-Time Exercise Questionnaire (GLTEQ), a simple, reliable, and effective tool, was used to assess physical activity [25,26]. It estimates physical activity over a typical week and is divided into three categories of intensity expressed as METs (Metabolic equivalent of task). The questionnaire score is expressed in units (strenuous > 24, moderate 14–23, mild < 14).

Quality of life (QoL) was evaluated with the DISABKIDS questionnaire, a validated questionnaire completed by children and their parents [27]. The modules used for the current study were the DISABKIDS chronic generic measure (DCGM-37) and the DISABKIDS disease-specific module for cystic fibrosis. The DISABKIDS chronic generic module (DCGM-37) consists of 37 items assigned to six dimensions: independence, emotional health, social inclusion, social exclusion, limitation, and treatment. These six dimensions can be combined to assess a general score for health-related QoL. The disease-specific CF questionnaire (DISABKIDS Cystic Fibrosis Module) consists of two domains: the impact domain (six items) on limitations and symptoms and the treatment domain (eight items) on treatment limitations related to CF.

### 2.4. Lung Function Testing

Spirometry was performed according to American Thoracic Society (ATS)/European Respiratory Society (ERS) guidelines [28]. Forced vital capacity (FVC) and Forced Expiratory Volume in 1 s (FEV_1_) were measured with a spirometer (Vitalograph 2120, Vitalograph Ltd., Ennis, Ireland). Data were expressed in percentage predicted values (pp) using normative data from the Global Lung Function Initiative software (GLI 2012, Global Lung Function Initiative Task Force, available at: http://www.lungfunction.org/ (accessed on 20 October 2022).

Multiple Breath Washout (MBW) was performed with a flow, volume, and molecular mass measurement analyzer (EXHALYZER D, Ecomedics, Switzerland) according to ATS/ERS guidelines [29]. 

### 2.5. Statistical Analysis

Descriptive and inferential data processing techniques were used for the statistical analysis. The uniform distribution of variables was tested. More specifically, statistical position and dispersion measures, frequency tables, and correlation coefficients were exported, while statistical significance tests on correlation coefficients, variance tests on means (ANOVA), and *t*-tests were performed. A suitable type of statistical analysis was implemented based on the variables used to extract each result.

The correlation between normally distributed and skewed continuous variables was assessed with the Student’s *t*-test and the Mann–Whitney test, respectively, with a categorical variable consisting of two groups. The correlation between the different parameters was evaluated with the Pearson correlation coefficient. A probability value of 5% was considered statistically significant (*p* < 0.05). Statistical analysis was performed using SPSS for Windows (IBM SPSS Statistics for Windows, Version 20.0. Armonk, NY, USA: IBM Corp.). The patients’ characteristics data were expressed as mean ± SD.

## 3. Results

Demographics of the CF group and age-matched healthy controls are shown in Table 1. The mean age of the CF group was 13.22 (±4.6), 49% boys, BMI, kg/m^2^ (19.58 ± 4.1), FVCpp (97.1% ± 12.4), and FEV1pp (99.7% ± 12.43), and LCI (10.16 ± 2.76). Mean BMI, weight, and height were comparable among the two groups (Table 1).

According to the GODIN physical activity questionnaire, 68% of the CF group were active, 27% were moderately active, and 5% had a more sedentary lifestyle, whereas 83% of the control group were active and 17% were moderately active, as shown in Figure 1. The mean GODIN score and the total QoL score were significantly lower among the CF group than the healthy controls, as shown in Figure 2.

PwCF were more independent compared to their healthy peers; while they presented a more affected QoL in total and experienced more physical restriction due to the disease, they also experienced a negative impact on their emotional health and social exclusion (stigma, feeling left out) (*p* < 0.05, Figure 2).

Moreover, a positive correlation between physical activity level and emotional health was observed (r^2^: 0.414, *p* < 0.05, Figure 3). Moreover, patients with more significant physical activity levels showed a better overall QoL (r^2^: 0.372, *p* < 0.05, Figure 4).

No significant correlation was found between PA and lung function parameters, FEV1pp, and LCI among pwCF (*p* = 0.572 and *p* = 0.271, respectively).

## 4. Discussion

In the present study, we evaluated the level of physical activity (PA) and its correlation with QoL among young CF patients. The results demonstrated that although most patients were active, they showed lower levels of PA than their healthy peers. The most affected QoL parameters among pwCF were emotional health, physical restrictions, and social exclusion. Furthermore, our study revealed a strong positive correlation between high levels of physical activity and both emotional health and overall QoL among pwCF.

Although it is well-established that physical activity benefits patients with CF, motivation and compliance remain challenging for children, adolescents, and young adults [30]. Patients are recommended to exercise regularly, with global PA guidelines of 150 min moderate-vigorous PA (MVPA) per week for adults and 60 min MVPA daily for children [29]. Tiredness, lack of time, and parental stress are the most frequently reported barriers [30]. Most of the young CF patients who participated in our study performed strenuous exercise (68%), several patients showed medium-intensity activity (27%), and only 5% showed a sedentary way of life. The CF study population presented with mild disease (mean FEV1 99.7% and mean BMI comparable to their age-matched healthy piers). Advances in CF treatment have contributed to better respiratory and growth outcomes among children with CF.

On the other hand, the healthy subjects who participated in the study were more active, with 83% showing strenuous physical activity and 17% presenting medium-intensity activity. Supervised training is more efficient in long-term adherence to the exercise program than an unsupervised or partially supervised exercise program [31,32]. Furthermore, advances in technology have contributed to the engagement of patients with CF in exercise training, as López-Liria et al. demonstrated that video games can be an excellent alternative to conventional treatment and recorded high levels of adherence to active components, making the treatment more entertaining [33].

The questionnaire used to evaluate the different variables that comprise the quality of life of the CF patients themselves was the DISABKIDS questionnaire. From the processing of the questionnaire results, it emerged that patients place particular emphasis on independence and social participation as factors that positively influence the QoL. At the same time, physical challenges and emotional distress are considered factors that can affect the patients’ quality of life. On the contrary, the group of healthy participants was described as positively influencing the quality of life, the feeling, and physical condition. The standard variable for a positive influence on the quality of life in the two groups was independence. This contrast in the assessment of physical activity is attributable to the reduced ability of individuals to pursue a vigorous exercise program and to the neglect observed in previous years regarding the inclusion of physical activity as part of the treatment.

In recent years, physicians caring for cystic fibrosis have suggested exercise as part of the treatment to improve lung function and respiratory capacity in children and adults. Among children with respiratory disease, regular exercise has been proven beneficial to their cardiovascular fitness, strength, and QoL [34]. Aerobic exercise training has a positive effect on exercise capacity, as well as on cardiovascular endurance and mucus clearance [7,35,36]. On the contrary, resistance training increases muscle strength and local muscle endurance [37]. The combination of both training modalities adapted to the patient’s age, and health status is recommended as the most profitable option [38]. Physical activity should always be associated with appropriate physiotherapy for successful respiratory recovery [39]. However, the frequency, intensity, and type of exercise required for health benefits are still unclear. In a CF group aged 2 to 40 years, Hebestreit et al. showed that high levels of exercise were correlated with a high aerobic capacity [40]. Exercise at moderate intensity is vital in maintaining health status, as it prevents muscle strength deterioration following hospitalization after a respiratory exacerbation in an adult [41]. In addition, sports participation can improve CF patients’ self-confidence and self-esteem [42,43]. This is supported by previous studies [44,45] in which patients who follow an exercise program of daily aerobic activity and strength-building exercises developed and improved their health-related QoL. Physical activity can be affected by several factors [46,47]. Especially in young people with CF, the burden of lung disease can affect their physical activity levels, which is an essential part of their socialization and group integration [15]. Due to lung disease, such individuals cannot participate in daily activity, and their emotional health and quality of life are negatively affected [37,48,49].

Our study has some limitations; the limited study group is one limitation of our study. Moreover, physical activity was assessed with a questionnaire; more sensitive and objective ways to assess physical activity have been established, like accelerometers. On the other hand, using questionnaires is a valuable way of evaluating physical activity and QOL. Mild lung disease (high FEV1pp) might also affect the negative correlation between physical activity and lung function. Moreover, our CF study group presented with mild lung disease and good nutritional status, which might have contributed to good physical activity and QOL scores; further progression of lung disease might affect the PA and QOL of CF patients.

The most important finding of the current study is that regular exercise or increased physical activity correlates with emotional health and overall QoL of pwCF. Selvadurai et al. [43] showed that aerobic exercise training improves patients’ QoL. In addition, according to the study by Klijn et al. [50,51], there is a high correlation between aerobic and anaerobic activity and increased quality of life of cystic fibrosis patients. Therefore, although significant benefits of regular workouts have already been observed in the quality of life of patients with cystic fibrosis, it becomes clear that there is an urgent need for further research into the effect of increased physical activity on health-related quality of life to reach certain conclusions and subsequently implement them in their treatment.

## Figures and Tables

**Figure 1 children-09-01665-f001:**
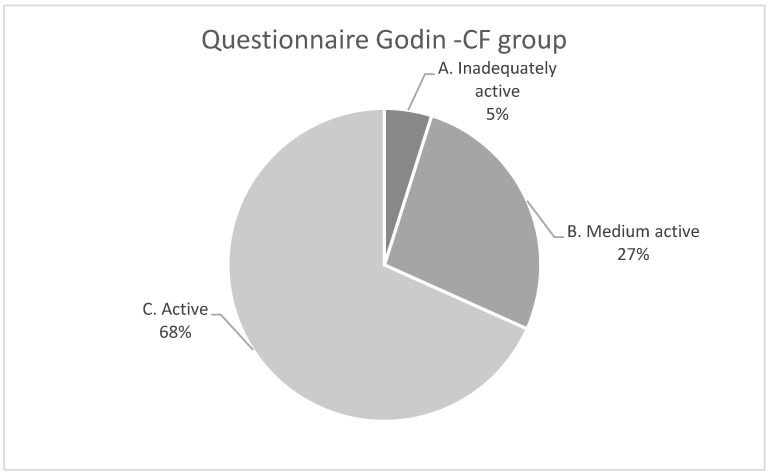
Physical activity (Godin Questionnaire) among pwCF and healthy controls.

**Figure 2 children-09-01665-f002:**
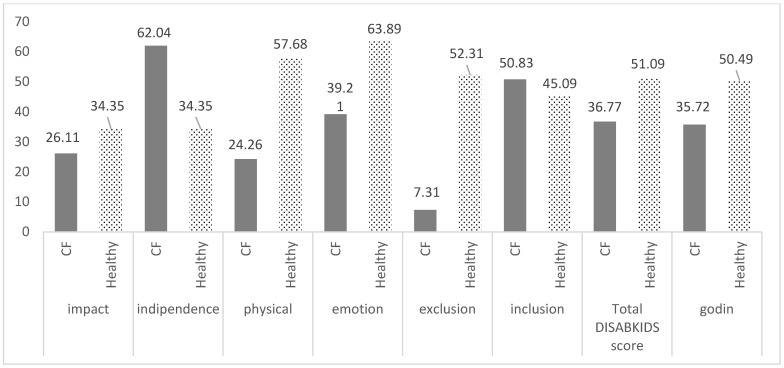
Healthy and patient participants’ quality of life (DISABKIDS questionnaire) and physical activity (Godin Q).

**Figure 3 children-09-01665-f003:**
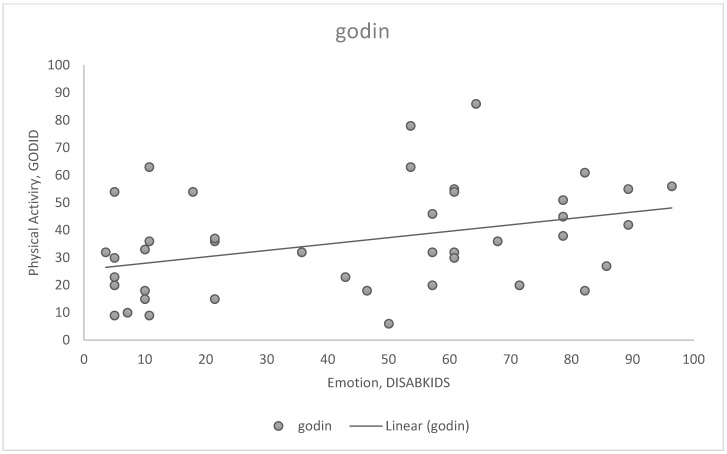
Patients with higher physical activity showed better emotional health (r: 0.414, *p*: 0.006).

**Figure 4 children-09-01665-f004:**
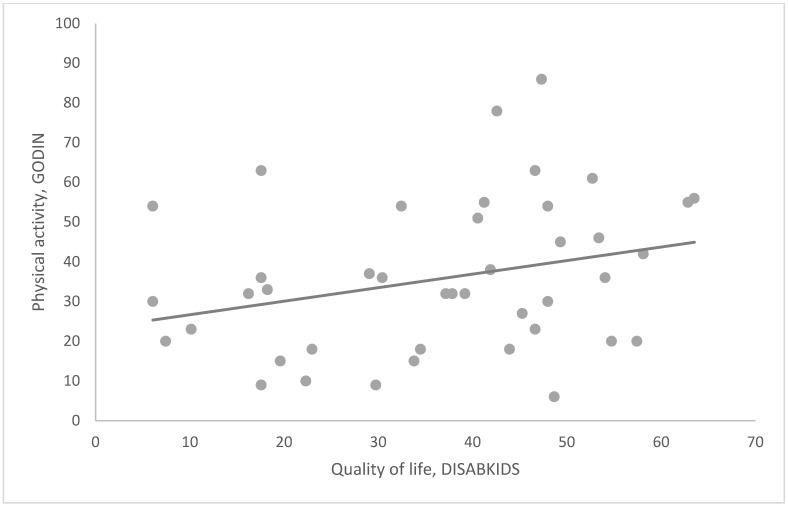
Patients with a greater degree of physical activity showed a better overall quality of life (r: 0.372; *p*: 0.014).

**Table 1 children-09-01665-t001:** Demographic characteristics of the study groups (pwCF and controls).

	Cystic Fibrosis (*n* = 45)	Healthy Control (*n* = 45)	*p*
**Age, years**	13.22 ± 4.6	13.80 ± 4.5	0.634
**ΒΜΙ (kg/m^2^)**	19.58 ± 4.1	19.57 ± 4.2	0.678
**Weight, kg**	44.47 ± 15.1	42.76 ± 15.3	0.670
**Height, cm**	149.22 ± 15.4	142.16 ± 15.7	0.156

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
