# Peer review of "Physical Activity and Quality of Life among Patients with Cystic Fibrosis"

_children, 2022, doi:10.3390/children9111665_

Round 1
Reviewer 1 Report
In this interesting study the authors tried to assess physical activity (PA) and quality of life (QoL) among children with cystic fibrosis compared to healthy controls; their secondary aim was to evaluate the correlation between PA and QoL on the one hand, and lung function (FEV1), on the other.
The overall manuscript is easy to flow, and I have only a few minor comments.
The authors start the discussion with the statement that “In the present study, we evaluated the level of physical activity (PA) and its impact on the QoL
I don’t think they measured the “impact”. Rather it should be the correlation between the two
The authors claim that “The most important finding of the present study is that regular exercise or increased physical activity can positively influence the emotion and overall quality of life of cystic fibrosis patients”
Again here- this statement has to be proven. Only a correlation was found and but not a casualty.
I agree that “more sensitive ways of assessing physical activity have been established, like using accelerometers…”, however, I’d add “more sensitive and OBJECTIVE ways….”
Author Response
We thank the reviewer for the valuable comments.
Please, find below our responses to your comments.
Reviewer 1: The authors start the discussion with the statement that “In the present study, we evaluated the level of physical activity (PA) and its impact on the QoL. I don’t think they measured the “impact.” Rather it should be the correlation between the two.
Author response: we thank the reviewer for the comment. We changed our statement accordingly.
Reviewer 1: The authors claim that “The most important finding of the present study is that regular exercise or increased physical activity can positively influence the emotion and overall quality of life of cystic fibrosis patients”. Again here- this statement has to be proven. Only a correlation was found and but not a casualty.
Author response: thank you. we changed our statement accordingly.
Reviewer 1: I agree that “more sensitive ways of assessing physical activity have been established, like using accelerometers…”; however, I’d add “more sensitive and OBJECTIVE ways….”
Author response: Thank you. We have changed the limitations, accordingly.
Reviewer 2 Report
The author has presented work regarding the analysis focused on “to evaluate
physical activity and quality of life among children with cystic fibrosis compared to healthy controls”.
The work is original and methodology is good. The discussion and conclusion are supported by the data.
I consider that it contains interesting information.
I had some comments as below.
Major:
1) In Table 1, there is no difference in height, weight, and BMI between patients and healthy subjects. And, the patients’ pulmonary function are within normal limits. Maybe, in this study, patients with mild symptoms without nutritional or respiratory disorders are mainly enrolled. The selection of cases should be described. A little more detailed patient characteristics should be described. And, the influence of this on the results of the study should be described in discussion.
2) If possible, I think you should mention the PA and QOL trends of patients with chest X-ray findings and/or respiratory symptoms.
If possible, I think you should mention the PA and QOL trends of patients with gastrointestinal symptoms.
3) In addition, I think it would be good to discuss the expected changes in PA and QOL when the disease stage progresses in the future.
4) The authors conclude that "Physical activity can positively influence the emotion and overall quality of life of cystic fibrosis patients." Is it possible that this was simply a result of feeling better because milder patients could have more PA? Even if it is just a guess, it should be stated how this PA and QOL will be affected by the difference in severity of CF.
Minor:
1) P1/L11 “CF”
Abbreviation “CF” for “cystic fibrosis” in abstracts should be fully spelled out on first appearance. I think that would be kinder for journals that are not limited to respiratory diseases.
2) P1/L12 “children with cystic fibrosis
“children with cystic fibrosis” can be abbreviated as “children with CF”.
3) P3L145 “13,22” Is this “,” correct?
And also L146 ”19,58” and L146 ”10,16”. And also in Table 1, “13,22” “13,80”…..
4) Have you evaluated CFTR variants?
Author Response
We would like to thank the reviewer for the valuable comments on our manuscript.
Please, find below our response to the reviewer's comments:
Reviewer comment 1: In Table 1, there is no difference in height, weight, and BMI between patients and healthy subjects. And, the patients’ pulmonary function are within normal limits. Maybe, in this study, patients with mild symptoms without nutritional or respiratory disorders are mainly enrolled. The selection of cases should be described. A little more detailed patient characteristics should be described. And, the influence of this on the results of the study should be described in discussion.
Author's response: we that the reviewer for the comments. We have added our response to the results section and we have added the relevant comment in the discussion section.
Reviewer comment 2: If possible, I think you should mention the PA and QOL trends of patients with chest X-ray findings and/or respiratory symptoms.
If possible, I think you should mention the PA and QOL trends of patients with gastrointestinal symptoms.
Author's response: Thank you for the comment; unfortunately we do not have the relevant data.
Reviewer comment 3: In addition, I think it would be good to discuss the expected changes in PA and QOL when the disease stage progresses in the future.
Author's response: Thank you; we have added the relevant comment in the discussion section.
Reviewer comment 4: The authors conclude that "Physical activity can positively influence the emotion and overall quality of life of cystic fibrosis patients." Is it possible that this was simply a result of feeling better because milder patients could have more PA? Even if it is just a guess, it should be stated how this PA and QOL will be affected by the difference in severity of CF.
Author's Response: Thank you; we have added the relevant comment in the discussion section.
Minor Comments:
R1) P1/L11 “CF”
Abbreviation “CF” for “cystic fibrosis” in abstracts should be fully spelled out on first appearance. I think that would be kinder for journals that are not limited to respiratory diseases.
Author response 1: thank you; we have corrected that.
R2) P1/L12 “children with cystic fibrosis
“children with cystic fibrosis” can be abbreviated as “children with CF”.
Author response: thank you; we corrected that.
R3) P3L145 “13,22” Is this “,” correct?
And also L146 ”19,58” and L146 ”10,16”. And also in Table 1, “13,22” “13,80”…..
Author response: thank you; we corrected that to ".".
R4) Have you evaluated CFTR variants?
Author response: thank you for the valuable comment; we have not evaluated CFTR variants.